# Genome-Wide Identification and Expression Analysis of WRKY Transcription Factors in *Akebia*
*trifoliata*: A Bioinformatics Study

**DOI:** 10.3390/genes13091540

**Published:** 2022-08-26

**Authors:** Jun Zhu, Shengfu Zhong, Ju Guan, Wei Chen, Hao Yang, Huai Yang, Chen Chen, Feiquan Tan, Tianheng Ren, Zhi Li, Qing Li, Peigao Luo

**Affiliations:** 1Provincial Key Laboratory for Plant Genetics and Breeding, Chengdu 611130, China; 2College of Agronomy, Sichuan Agricultural University, Chengdu 611130, China; 3Sichuan Akebia trifoliata Biotechnology Co., Ltd., Chengdu 611130, China; 4Department of Biology and Chemistry, Chongqing Industry and Trade Polytechnic, Chongqing 408000, China

**Keywords:** *WRKY* gene, *Akebia trifoliata*, genome duplication, transcriptome analysis, disease resistance

## Abstract

WRKY transcription factors have been found in most plants and play an important role in regulating organ growth and disease response. Outlining the profile of *WRKY* genes is a very useful project for studying morphogenesis and resistance formation. In the present study, a total of 63 *WRKY* genes consisting of 13 class I, 41 class II, and 9 class III genes were identified from the newly published *A. trifoliata* genome, of which 62 were physically distributed on all 16 chromosomes. Structurally, two *AkWRKY* genes (*AkWRKY6* and *AkWRKY52*) contained four domains, and *AkWRKY17* lacked the typical heptapeptide structure. Evolutionarily, 42, 16, and 5 *AkWRKY* genes experienced whole genome duplication (WGD) or fragmentation, dispersed duplication, and tandem duplication, respectively; 28 Ka/Ks values of 30 pairs of homologous genes were far lower than 1, while those of orthologous gene pairs between *AkWRKY41* and *AkWRKY52* reached up to 2.07. Transcriptome analysis showed that many of the genes were generally expressed at a low level in 12 fruit samples consisting of three tissues, including rind, flesh, and seeds, at four developmental stages, and interaction analysis between *AkWRKY* and *AkNBS* genes containing W-boxes suggested that *AkWRKY24* could play a role in plant disease resistance by positively regulating *AkNBS18*. In summary, the *WRKY* gene family of *A. trifoliata* was systemically characterized for the first time, and the data and information obtained regarding *AkWRKY* could be very useful in further theoretically elucidating the molecular mechanisms of plant development and response to pathogens and practically improving favorable traits such as disease resistance.

## 1. Introduction

Transcription factors (TFs) play a non-negligible role in plant organ development and disease resistance [1,2,3]. Plants contain many types of TFs, such as WRKY, bHLH (basic/helix–loop–helix), ZF-HD (zinc finger homeodomain protein), MYB (myeloblastosis), and NAC (NAM, ATAF, CUC) [4]. Among these, the WRKY TFs comprise one of the largest families in plants and are named after the highly conserved WRKYGQK sequence [5]. Based on the number of WRKY domains and the type of zinc finger structure, the WRKY family can be divided into three categories. Group I genes possess two WRKY domains with a C2H2 zinc finger structure; group II genes possess a WRKY domain with a C2H2 zinc finger structure; and group III genes possess a WRKY domain with a C2HC zinc finger structure. In addition, group II can be further divided into five subgroups, IIa, IIb, IIc, IId, and IIe, according to phylogenetic data [6,7].

Studies have shown that WRKY TFs are not only widely involved in innate immunity in both pathogen-associated molecular patterns (PAMPs) and effector-triggered immunity (ETI) patterns, but also in the response to abiotic stresses such as drought, salt, and cold [8,9,10,11]. In addition, WRKY TFs are involved in growth and development processes such as seed dormancy and leaf senescence [12,13,14]. In fact, some *WRKY* genes could be pleiotropic. For example, *FvWRKY42* overexpression in *Arabidopsis* not only enhanced resistance to powdery mildew but also improved plant tolerance to both drought and salt stress [15]. The WRKY protein plays its role mainly by binding to the W-box (TTGACT/C) in the promoter region of the target gene, in which TGAC is an invariant core sequence and thus crucial for the function of the WRKY protein [16,17,18]. Therefore, we can determine whether the gene is directly regulated by the WRKY transcription factor according to the presence or absence of the W-box.

*Akebia trifoliata*, commonly known as August melon or wild banana, belongs to the perennial woody liana plant family and grows in Asia, especially China, Japan, and Korea [19]. *A. trifoliata* has been used as an effective traditional Chinese medicine for diuretic, anti-inflammatory, ulcer-care, and indigestion-relief applications [20,21,22]. In addition, some authors have found that extracts of the fruits and stems of *A. trifoliata* can inhibit the proliferation of liver cancer and gastric cancer cells, respectively [23,24]. Another economically important application of *A. trifoliata* is as an edible fruit, which is very rich in sugars, proteins, vitamins, saponins, and free amino acids. Additionally, the content of minerals such as iron, zinc, calcium, manganese, and magnesium is significantly higher in *A. trifoliata* fruits than in common fruits such as apples and pears [25,26]. It is evident that *A. trifoliata* has great prospects for use as both a medicinal plant and a fruit crop.

The WRKY family has been extensively studied in many crops, such as rice, grape, soybean, and barley [27,28,29,30], but no report has been published on the WRKY family in *A. trifoliata*. To accelerate the theoretical study and commercial exploitation of *A. trifoliata*, it is of great significance to systematically analyze the WRKY family. The public genome data recently uploaded by our team and the reported transcriptomic data of *A. trifoliata* [31] provide an opportunity to characterize the structural, evolutionary, and functional profile of the WRKY family. In this paper, the structure, conserved motifs, chromosome localization, homology, and expression patterns of 63 *WRKY* genes from *A. trifoliata* are described in detail. The possible downstream target *NBS* genes were predicted according to the presence of the W-box and the correlation analysis between *WRKY* and *NBS* genes, which provided valuable clues for revealing the disease resistance of *A. trifoliata*.

## 2. Materials and Methods

### 2.1. Data Used in This Study

Due to the unavailability of the corresponding assembled files, the first published genome of *A. trifoliata* subsp. *Australis* [32] was not used in the present study. Instead, we used genomes from another BioProject (ID PRJNA671772) of the National Center for Biotechnology Information (NCBI) (https://www.ncbi.nlm.nih.gov/bioproject?LinkName=assembly_bioproject&from_uid=9862971; accessed on 30 November 2021) and Big Data Center accession number GWHBISH00000000 (https://ngdc.cncb.ac.cn/; accessed on 25 February 2022). The raw data and the related files of the transcriptomic data of 12 samples consisting of three tissues (rind, flesh, and seed) at four stages (young, enlargement, coloring stage, and mature stage) have recently been reported [31], and they were further used to outline the expression profile of *WRKY* genes (https://www.ncbi.nlm.nih.gov/sra/?term=Akebia+trifoliata; SRA numbers: SRX9395000-SRX9395009, SRX9395011-SRX9395036; accessed on 14 December 2021).

### 2.2. Identification of Akebia trifoliata AkWRKY

To identify *WRKY* genes from the *A. trifoliata* genome, 125 and 75 reference *WRKY* genes of rice and *Arabidopsis*, respectively, were used as query sequences to perform local BLASTp with an E-value of 1e-10 [5,33]. The hidden Markov model (HMM) of the WRKY domain (PF03106) in the Pfam database (http://pfam.sanger.ac.uk/, accessed on 5 January 2022) was further used to search for conserved domains among the identified candidate sequences obtained from BLASTp. Additionally, the conserved domain was analyzed using the conserved domain database (CDD) in NCBI (https://www.ncbi.nlm.nih.gov/Structure/bwrpsb/bwrpsb.cgi, accessed on 13 January 2022), and the domain prediction results were visualized by TBtools software [34]. The conserved motif was identified using MEME (https://meme-suite.org/meme/doc/meme.html, accessed on 20 January 2022).

### 2.3. Protein Properties of AkWRKY

Subsequently, the amino acid sequences of AkWRKYs were retrieved from the *A. trifoliata* genome and used to analyze physical and chemical properties, and subcellular locations were predicted using ProtParam (http://web.expasy.org/protparam/, accessed on 30 January 2022), TMHMM Server v. 2.0 (http://www.cbs.dtu.dk/services/TMHMM-2.0/, accessed on 30 January 2022), SignalP 4.1 (http://www.cbs.dtu.dk/services/SignalP/, accessed on 1 January 2022), WoLF PSORT (http://www.genscript.com/wolf-psort.html, accessed on 2 January 2022), and UniProt (https://www.uniprot.org/, accessed on 2 January 2022). The motif composition was analyzed using the Multiple Em for Motif Elicitation (MEME) online program (http:/meme.nbcr.net/meme/intro.html, accessed on 4 January 2022) [35].

### 2.4. Phylogenetic Analysis

The amino acid sequences of the reference WRKYs from *Arabidopsis* were obtained from the Ensembl database (http://plants.ensembl.org/info/data/ftp/index.html, accessed on 8 January 2022) and used for phylogenetic analysis against 63 *A. trifoliata* WRKYs. The phylogenetic tree was constructed using MEGA7.0 with the maximum likelihood (ML) method (http://www.megasoftware.net, accessed on 9 January 2022) and 1000 bootstrap replications.

### 2.5. Exon–Intron Structure Analysis

The cDNA and genomic sequences of *A. trifoliata WRKY* genes were obtained from NCBI and then used to analyze the exon–intron organizations using the Gene Structure Display Server (GSDS) (http://gsds.cbi.pku.edu.cn/index.php, accessed on 12 January 2022).

### 2.6. Chromosomal Location, Gene Replication, and Ka/Ks Analysis

The physical location of *A. trifoliata WRKY* genes was obtained from position information in the GFF3 file. Subsequently, they were mapped to *A. trifoliata* chromosomes using Circos [36]. The gene replication events among *A. trifoliata WRKY* genes were analyzed using multiple collinear scanning toolkits (MCScanX) with the default parameters [37]. Subsequently, Calculator 2.0 was used to calculate the Ka/Ks ratio of the identified *WRKY* genes [38].

### 2.7. AkWRKY Gene GO Annotation and KEGG Annotation

TBtools was adopted to visualize the annotated results using the Blast2 GO tool with default parameters [39]. The *WRKY* genes of *A. trifoliata* were annotated in the KEGG database website and then mapped and analyzed with TBtools, and the KEGG pathway map was drawn using the online KEGG mapping tools (https://www.kegg.jp/kegg/mapper/reconstruct.html, accessed on 17 January 2022).

### 2.8. Expression Pattern of AkWRKY Genes in A. trifoliata Fruit Tissues

First, RNA-seq data of 12 samples were aligned with the genome of *A. trifoliata* [19]. Second, the SAM tool was used to compress the alignment results into BAM format files, and the expression values of the *WRKY* genes were extracted according to a previously reported method [40]. Finally, the expression data were made into a heatmap with the help of TBtools [34].

### 2.9. Putative Promoter Region Analysis

To investigate the cis-acting elements of the putative promoter regions, the W-box within 2000 bp upstream of the start codon of 73 *AkNBS* genes and 63 *AkWRKY* genes was analyzed using the online site PlareCARE (http://bioinformatics.psb.ugent.be/webtools/plantcare/html/, accessed on 19 January 2022).

### 2.10. Correlation Analysis between AkWRKY Genes and AkNBS Genes

To further study the correlation between *WRKY* genes and *NBS* genes, 18 *NBS* genes with W-box and 63 *WRKY* genes were analyzed by the R package Corrplot (https://github.com/taiyun/corrplot, accessed on 25 January 2022).

## 3. Results

### 3.1. Identification and Classification of the AkWRKY Genes of A. trifoliata

A total of 63 *WRKY* genes were identified in *A. trifoliata*; 62 *WRKY* genes were named *AkWRKY1* to *AkWRKY62* according to their chromosomal locations (Figure 1), and the remaining gene in the unassembled contig was named *AkWRKY63* (Appendix A). The average total length and CDS length were 3854 and 1161, ranging from 698 to 12,174 and from 396 to 3318, respectively (Appendix A). Structurally, all 63 *WRKY* genes were interrupted by introns, and the number of exons ranged from two to seven (Figure 2). A total of 32 out of the 63 *WRKY* genes (50.8%) contained three exons, and *AkWRKY6* and *AkWRKY52* had the most (seven exons), while *AkWRKY18*, *AkWRKY38* and *AkWRKY51* had the least (two exons) (Appendix A). Out of the 63 *AkWRKYs* (98.4%), 62 were distributed on all 16 chromosomes of *A. trifoliata*, of which most were located in the chromosomal end regions (Figure 1). The number of *AkWRKY* genes distributed on each chromosome ranged from one to seven. Chromosomes 1 and 11 contained the mostseven *AkWRKY* genes, while chromosome 9 contained only one *AkWRKY* gene.

According to the phylogenetic tree (Figure 3), the 63 *AkWRKY* genes were divided into three groups, I, II and III, containing 13, 41, and 9 *AkWRKY* genes, respectively. Group II was further divided into five subgroups: IIa (3), Iib (8), Iic (19), Iid (4), and IIe (7). Further comparison analysis found that group I *AkWRKY* genes had a significantly larger gene length and CDS length than both groups II and III genes at the *p* = 0.05 level (Table 1 and Appendix A).

### 3.2. Basic Information and Motif Composition of the AkWRKY Protein

Basic information, including the number of amino acids (NA), isoelectric point (PI), molecular weight (MW), and subcellular localization of the proteins encoded by 63 *AkWRKY* genes, is listed in Appendix A. The average NA, MW, and PI were 386, 42,942.85 kDa, and 6.95, with variations from 131 (*AkWRKY55*) to 1105 (*AkWRKY55*), from 14,971 (*AkWRKY55*) to 120,803 (*AkWRKY52*), and from 4.90 (*AkWRKY50*) to 9.70 (*AkWRKY20*), respectively. Most of these genes (59) were located in the nucleus, while only one existed in the cytoplasm (*AkWRKY11*), chloroplast (*AkWRKY35*), peroxisome (*AkWRKY38*), and vacuole (*AkWRKY56*), respectively. In addition, only two AkWRKY proteins (AkWRKY6 and AkWRKY52) contained four WRKY domains, while 51 of the 63 AkWRKY proteins contained only one WRKY domain, and the remaining ten AkWRKY proteins contained two WRKY domains (Appendix A).

The conserved motifs of the AkWRKY protein were further predicted using MEME (Appendix A), and the results showed that motifs 1 and 2 could be hallmarks of typical WRKY highly conserved sequences and zinc finger sequences, respectively. Almost all WRKY proteins contained motifs 1 and 2. Although AkWRKY17 lacked motif 1, the WRKY domain contained in the gene was confirmed by SMART analysis. The distribution of different motifs also showed subfamily specificity. For example, all 13 group I AkWRKY proteins except AkWRKY40 contained motifs 3 and 5, while motif 6 only existed in group IIa or IIb (Appendix A). In addition, 7 of the 63 AkWRKY proteins had four forms of heptapeptide variants: WRKYGKK (AkWRKY1, AkWRKY8, AkWRKY11, and AkWRKY13); WCKYGRK (AkWRKY6 and AkWRKY52); WKKYGQK (AkWRKY55); and WLKYGKK (AkWRKY52) (Appendix A). Interestingly, the four heptapeptides of AkWRKY52 consisted of two WRKYGQKs, one WCKYGRK, and one WLKYGKK.

### 3.3. Duplication and Natural Selection Type of AkWRKY Genes

MCScanX analysis showed that five *AkWRKY* genes experienced tandem duplication, 42 experienced whole genome duplication (WGD) or fragmented duplication, and the remaining 16 *AkWRKY* genes experienced dispersed duplication (Appendix A). There were four duplicated genes on chromosome 4, while chromosomes 2, 3, and 15 did not contain WGD or fragment duplication. In addition, we detected a total of 30 paralogous pairs among the *AkWRKY* genes, and the calculated Ka/Ks values with an average of 0.41 ranged from 0.18 (*AkWRKY38* and *AkWRKY51*) to 2.07 (*AkWRKY41* and *AkWRKY52*). Among them, 28 (93.3%) calculated Ka/Ks values were far lower than 1, while only one (*AkWRKY5* and *AkWRKY52*) (Appendix A) was very close to 1, indicating that the *WRKY* genes in *A. trifoliata* were mainly selected for purification in the evolutionary process.

### 3.4. GO and KEGG Enrichment Analysis of AkWRKY Genes

The 63 *AkWRKY* genes were divided into three categories, molecular function, cell composition, and biological processes, by GO enrichment analysis, with 11, 5, and 125 subcategories, respectively (Appendix A). Fifty-five *AkWRKY* genes were involved in molecular functions, such as transcriptional regulatory activity and DNA-binding transcription factor activity; seventeen in cell composition; and fifty-five in biological processes, such as RNA biosynthesis, the regulation of cell metabolism, the regulation of biosynthesis, nucleic acid metabolism, heterocyclic biosynthesis, and transcription regulation. Likewise, they were also divided into five categories, plant–pathogen interactions, environmental adaptation, organic systems, TFs, and protein families (genetic information processing), by KEGG enrichment analysis, in which the plant–pathogen interaction category was ranked in the first position (Figure 4).

### 3.5. Transcriptome Analysis of AtWRKY Genes in Different Tissues of A. trifoliata Fruit

The analysis of a transcriptomic dataset consisting of 12 samples including three tissues (rind, flesh, and seed) at four developmental stages (youth, enlargement, coloring, and maturity) of *A. trifoliata* fruit showed that all 63 *AkWRKYs* were expressed at detectable levels, although most of them were expressed at low levels, and that their expression level was generally higher in the late developmental stages than in the early developmental stages (Figure 5A). Overall, *AkWRKY37* showed the highest expression, while *AkWRKY38* exhibited the lowest expression in fruit tissues. *AkWRKY4*, *7*, and *16* were significantly expressed in the rind and their expression levels increased with time. *AkWRKY12*, *24*, *27*, and *37* all had the highest expression levels in the third stage of rind development. *AkWRKY7* and *37* not only had high expression in the rind, but also had relatively obvious expression in the flesh. *AkWRKY4*, *7*, *12*, *16*, *24*, *27*, and *37* had the largest difference between the lowest and highest expression levels (Appendix A). The different expression characteristics of different *AkWRKY* genes in the fruit tissues of *A. trifoliata* indicated that this family may be involved in a wide range of developmental pathways, playing different roles. In addition, the average expression levels of *WRKY* genes in the rind, flesh, and seed were 12.15, 4.18, and 4.51, respectively, and the expression levels of *WRKY* genes in the peel were approximately three times higher than those in both the pulp and seed (Figure 5B). In addition, some *AkWRKY* genes in the same subfamily had similar expression patterns, such as *AkWRKY12* and *AkWRKY37* in the IId group.

### 3.6. Putative Downstream Target WRKY and NBS Genes Regulated by WRKY Genes

To understand the downstream target *WRKY* and *NBS* genes regulated by the *AkWRKY* genes, the W-box cis-acting elements in 63 *AkWRKY* genes and 73 previously reported *AkNBS* genes were predicted by Yu et al. (2021) [41]. Among the 63 *AkWRKY* genes, 24 contained a W-box, of which 8 contained two W-boxes and 2 contained 3 W-boxes. Among the 73 *AkNBS*, 21 contained W-box cis-acting elements, of which 4 *AkNBS* genes (*AkNBS34*, *AkNBS41*, *AkNBS42*, and *AkNBS66*) contained 2 W-boxes and the remaining 17 contained only 1 W-box (Appendix A). Correlation analysis between 63 *AkWRKY* genes and 18 of the 21 W-box *AkNBS* genes was carried out because there was no detectable expression level of the remaining three W-box genes (*AkNBS16*, *AkNBS38*, and *AkNBS55*) in the same transcriptomic dataset, and the results are shown in Appendix A. Both *AkNBS* and four *AkWRKY* genes were selected for further correlation analysis according to FPKM and correlation values (Appendix A), and the results showed that there was a strong relationship between *AkNBS18* and *AkWRKY24* (Figure 6). Similarly, there were four strong correlation coefficients involving four *WRKY* genes, among which only *AkWRKY49* contained a W-box.

## 4. Discussion

### 4.1. The WRKY Gene Family of A. trifoliata Follows a Conservative Classification System

As a basal eudicot, *A. trifoliata* plays an important role in the early evolution of eudicots [42]. WRKY TFs are one of the largest families in plants and modulate many biological processes, especially disease resistance, which provides important insight into the evolutionary signaling webs of transcriptional regulators [43]. In the present study, the number of *WRKY* genes identified in the *A. trifoliata* genome was 63, which was more than that in both *Ostreococcus tauri* [44] and *Selaginella moellendorffii* [45], less than that in both *Pinus monticola* [46] and *Malus domestica* [47], and close to that in *Vitis vinifera* [29] (Appendix A). Various studies have suggested that both WRKY domains and zinc finger structures are evolutionarily highly conserved [6,43,48]; therefore, both the number of WRKY domains and the type of zinc finger structure are usually employed for classification. According to the classification of 75 reference *WRKY* genes of the *Arabidopsis* genome, all 63 identified *WRKY* genes were classified into three groups, I, II and III, among which group II consisted of five subgroups (Figure 3). The number of *WRKY* genes in group II was the highest (41, 65.1%), while that in group III was the lowest (9, 14.3%); the number distribution between the different groups in *A. trifoliata* was similar to that in most eudicots, such as *Santalum album*, *Vitis vinifera*, and *Daucus carota* [29,49] (Appendix A). Similarly, the IIc subgroup of group II contained 19 *AkWRKY* genes, accounting for 30.2% (i.e., the most members) of the total, while the IIa subgroup only contained three members, accounting for 4.8% (i.e., the fewest members) of the total (Figure 3). Comparison analysis found that the average number of exons, gene length, and CDS length of group I were significantly greater than those of groups II and III at the *p* = 0.05 level. In summary, along with the difference in the exon number, gene length, and CDS length of *WRKY* genes in different classifications, the uneven gene number in different groups/subgroups and uneven physical distribution on chromosomes (Figure 1 and Appendix A) indicated that different *WRKY* genes could have experienced different evolutionary events, such as genome duplication style and both the type and strength of natural selection.

### 4.2. The Variation in the Number and Components of Heptapeptides in WRKY Genes

Both the heptapeptide structure and the zinc finger structure are structurally important and functionally crucial characteristics of *WRKY* genes; therefore, motif 1, containing the conserved heptapeptide, and motif 2, containing the zinc finger structure, could be treated as hallmarks of *WRKY* genes. We found that all *AkWRKY* genes contained motif 2, almost all contained motif 1, and only *AkWRKY17* lacked motif 1 and a heptapeptide structure [5], which reinforced the view that heptapeptide and zinc finger structures are highly conserved. Compared with the conserved heptapeptide, the zinc finger tolerates less variation in both structure and number. In the present study, the zinc finger structure of all 63 *AkWRKY* genes comprised both normal C2H2 and C2HC types (Appendix A). In contrast, we found that both *AkWRKY6* and *AkWRKY52* had four heptapeptide structures, while *AkWRKY17* lacked this structure. We detected 78 heptapeptides on 63 *AkWRKY* genes. Among the 13 group I genes, 2 had four heptapeptides and 11 had two heptapeptides; however, only one heptapeptide was detected in the remaining *AkWRKY40*, but two WRKY domains could be identified by SMART (Appendix A), which suggested that the additional heptapeptides were mainly concentrated on group I *WRKY* genes in the *A. trifoliata* genome. We also found that eight (10.3%) mutants of heptapeptides putatively varied from the classic WRKYGQK. In addition, *AkWRKY52* had two mutant heptapeptides (Appendix A). The evidence suggested that although the heptapeptide structure was highly conserved in plant WRKY proteins, it could better tolerate certain variations in both number and sequence, especially in *WRKY* genes with additional heptapeptides.

In addition, we also found that both the position and number of the varied amino acids of the heptapeptides could be group- or subgroup-specific. For example, the second “R” or the sixth “Q” was transferred into “K” in the corresponding position on five (26.3%) of the 19 IIc group genes (Appendix A). The varied heptapeptides in both *AkWRKY6* and *AkWRKY52* contained the same two amino acid transitions at the second and sixth positions, from “R” into “C” and from “Q” into “R”, respectively, belonging to group I. In fact, previous reports on Populus, banana, and maize found similar results [50]. Finally, we noticed that *AkWRKY6* and *AkWRKY52*, which contained four heptapeptides, had significantly greater CDS lengths, molecular weights, and both exon and amino acid numbers (Appendix A), which indicated that the larger the gene, the greater the number of heptapeptides and that heptapeptide-coding segment duplication could be responsible for the increase in the number of heptapeptides in the evolutionary process.

### 4.3. WRKY Genes Evolutionarily Experienced Genome Duplication and Natural Selection Events

Previous studies have concluded that gene duplication events are among the important drivers of gene family expansion [51]. In this study, it was found that the repeat types of 63 *AkWRKY* genes were mainly WGDs or segment repeats, dispersed repeats, and tandem repeats, and there was no single copy. Of these, 42 (66.7%) were WGDs or fragment repeats, 16 (25.4%) were dispersed repeats, and 5 (7.9%) were tandem repeats (Appendix A). This showed that the WRKY family in *A. trifoliata* may be derived from the duplication of other genes. Obviously, although dispersed and tandem duplication partly contributed to the production of the *WRKY* gene family, WGD or fragmental genomic duplication could be the main evolutionary event experienced by *WRKY* genes in *A. trifoliata*.

Additionally, the Ka/Ks value is usually treated as an important informative indicator of both the type and the strength of natural selection [41]. The watershed Ka/Ks value between purifying selection and positive selection is 1.0, which indicates neutral selection, and if the calculated value is larger than 1.0, the corresponding orthologous gene pairs could have experienced positive selection; likewise, if the value is smaller than 1.0, they could have experienced purifying selection [52]. Among the 30 Ka/Ks values of the orthologous gene pairs, 28 were far lower than 1.0, one was close to 1.0, and one was far greater than 1.0, which suggested that many *AkWRKY* genes had experienced strongly purifying selection (Appendix A). Interestingly, we found that all three *AkWRKY* genes (*AkWARY5*, *AkWARY41*, and *AkWARY52*) that contributed to the three largest Ka/Ks values of the orthologous gene pairs (*AkWARY41* and *AkWARY52* with 2.07, *AkWARY5* and *AkWARY52* with 1.06, and *AkWARY5* and *AkWARY41* with 0.61) belonged to group I; thus, unlike those in groups II and III, different *AkWRKY* genes in group I could have experienced different types and strengths of natural selection.

### 4.4. Prediction of the Potential Downstream Target Genes of WRKY Genes in the Disease Resistance Process

The highly conserved W-box (TTGACC/T) is the cognate binding site of WRKY TFs, which consequently mirrors the conservation of the WRKY domain [6]. Therefore, the W-box is the minimal and essential consensus required for specific DNA binding [53], and almost all WRKY TFs preferentially bind to the W-box, although there have been a few reports of WRKY proteins binding to non-W-box sequences [43,54]. In fact, the WRKY proteins binding to non-W-box sequences had at least a transited amino acid sequence rather than the common WRKYGQK [55], which further reinforced rather than denied the view that the W-box could be a hallmark of the downstream target genes of *WRKY* genes. Obviously, bioinformatics would be a useful method for predicting the candidate target genes possibly regulated by WRKY TFs by identifying the existence of the W-box.

In the present study, the W-box was detected in 63 *AkWRKY* and 73 *AkNBS* genes because of the putative function of *AkWRKY* genes [43], the available information regarding *AkNBS* genes [41], and the urgency of *A. trifoliata* disease resistance improvement [56]. The results showed that 24 (38.1%) out of the 63 *AkWRKY* genes contained W-boxes, and the number of W-boxes ranged from one to three, with ten (41.7%) *AkWRKY* genes containing more than one W-box. In contrast, 21 (28.8%) out of the 73 *AkNBS* genes contained W-boxes. The largest number of W-boxes was two, though only four (19.0%) *AkNBS* genes (*AkNBS34*, *AkNBS41*, *AkNBS42*, and *AkNBS66*) contained two W-boxes (Appendix A). This granted us some valuable evidence to speculate that autoregulation and cross-regulation could exist extensively in many *AkWRKY* genes, and both the multiple regulating styles and various regulatory abilities of *AkWRKY* genes also afforded a reasonable explanation for the small number of *AkWRKY* genes (Appendix A). In addition, the expression characteristics of the *AkNBS* genes and *AkWRKY* genes containing W-boxes were the same: both were highly expressed in the pericarp, and their expression increased with the change in developmental process (Appendix A). Previous studies have demonstrated that the NBS family plays a very important role in the defense system of plant disease resistance, which is usually regulated by *WRKY* genes [57]. Both GO (Appendix A) and KEGG (Figure 4) enrichment analyses indicated that at least some *AkWRKY* genes could be involved in plant–pathogen interaction processes. Additionally, the 21 *AkNBS* genes containing W-boxes could be involved in the disease resistance response putatively regulated by *AkWRKY* genes.

### 4.5. AkWRKY24 Could Be Involved in the Disease Resistance Process Possibly by Regulating AkNBS18

Transcriptomic analysis is an important method for studying plant genes, especially resistance gene function [58,59]. We found that the expression level of all 63 *AkWRKYs* could be detected, although it was generally low, and that the expression level increased with fruit development (Figure 5A, Appendix A). Similarly, 66 (90.4%) of the 73 *AkNBS* genes had a detectable expression level, while 28 *AkNBS* genes had a higher expression level than the threshold for categorization as genes with low expression [41]. Subsequently, we examined the correlations between all 63 *AkWRKY* genes and 18 detectable-expression *AkNBS* genes with W-boxes, and the results showed that two *AkNBS* and four *AkWRKY* genes were correlated (Figure 6). We also found a strong relationship between *AkNBS18* and *AkWRKY24* (Figure 6).

In the present study, both *AkNBS18* and *AkWRKY24* exhibited high expression in rinds compared with flesh and seeds (Appendix A, which are the outermost parts exposed to various pathogens [60,61]. Second, *AkNBS18*, belonging to the CNL type, could be independently involved in the resistance response and did not need the help of RNL [62]. Third, *AtWRKY54*, as the homologous gene of *AkWRKY24* in group III, is involved in plant defense in *Arabidopsis* [63,64,65]. Hence, we propose that *AkWRKY24* could take part in the disease resistance process by regulating *AkNBS18*.

## 5. Conclusions

In conclusion, we identified 63 *WRKY* genes in the *A. trifoliata* genome. The *WRKY* genes were classified into three groups, I, II (with five subgroups), and III and were physically mapped on all 16 chromosomes. The conserved zinc finger and heptapeptide motifs were analyzed. Compared with the zinc finger structure, the heptapeptides showed some variations in both structure and number. WGD or fragmental genomic duplication and purifying selection were the main evolutionary events experienced by *WRKY* genes in *A. trifoliata*. Many *WRKY* genes had a W-box, so they could be functionally autoregulated and cross-regulated. Transcriptome data analysis showed that the genes of the AkWRKY family had multiple expression patterns, indicating that they may have multiple functions, while some *AkWRKY* genes and *AkNBS* genes had similar expression patterns, indicating that these genes may be functionally related. Finally, co-expression analysis between all *AkWRKY* genes and 18 W-box *AkNBS* genes suggested that *AkWRKY24* could take part in the disease resistance response, possibly by regulating *AkNBS18*. Therefore, this study added some important data to the plant WRKY transcription factor pool and afforded some new tools for studying the molecular mechanism of resistance in *A. trifoliata*.

## Figures and Tables

**Figure 1 genes-13-01540-f001:**
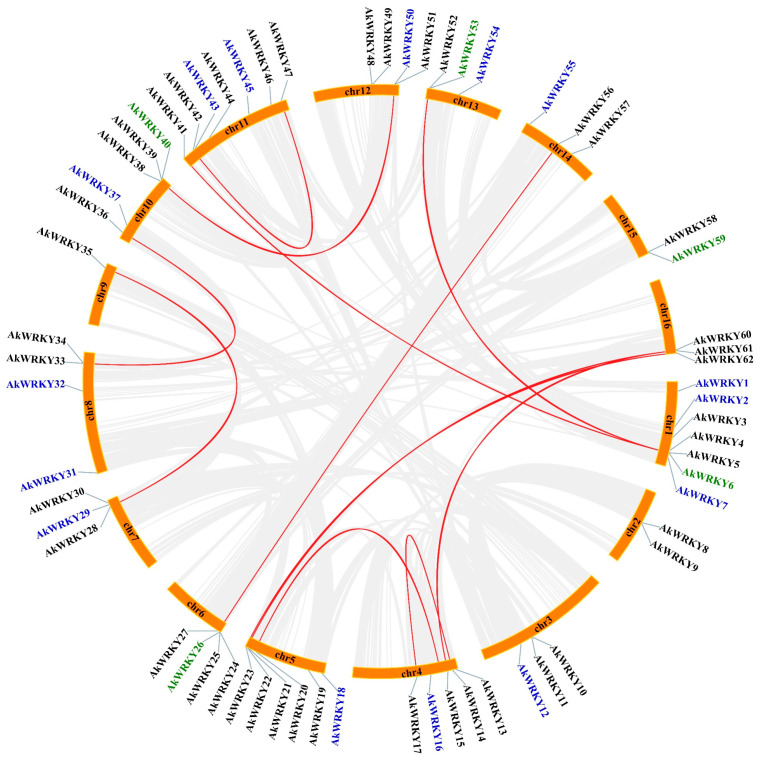
The chromosomal location of the *WRKY* genes in *A. trifoliata*. The red lines in the inner circle represent the gene pairs from fragment duplication. Genes marked in black are WGDs or fragment duplications, genes marked in blue are dispersed duplications, and genes marked in green are tandem duplications.

**Figure 2 genes-13-01540-f002:**
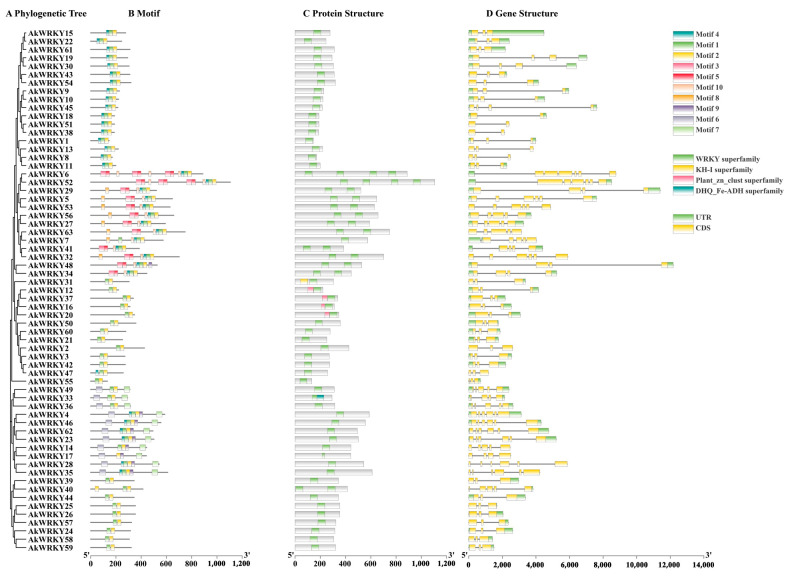
Gene structure and motif analysis of *WRKY* genes. (**A**) Phylogenetic tree of 63 *AkWRKY* genes. (**B**) By analyzing the amino acid sequence, 10 composition motifs were obtained, and motifs were drawn in different colors. (**C**) Schematic diagram of the domains of the 63 AkWRKY proteins; green indicates the characteristic WRKY domain. (**D**) The genetic structure of the *WRKY* genes, with exons separated by introns that are represented by thin lines. Light green indicates the untranslated 5’ and 3’ regions, and yellow indicates the exons.

**Figure 3 genes-13-01540-f003:**
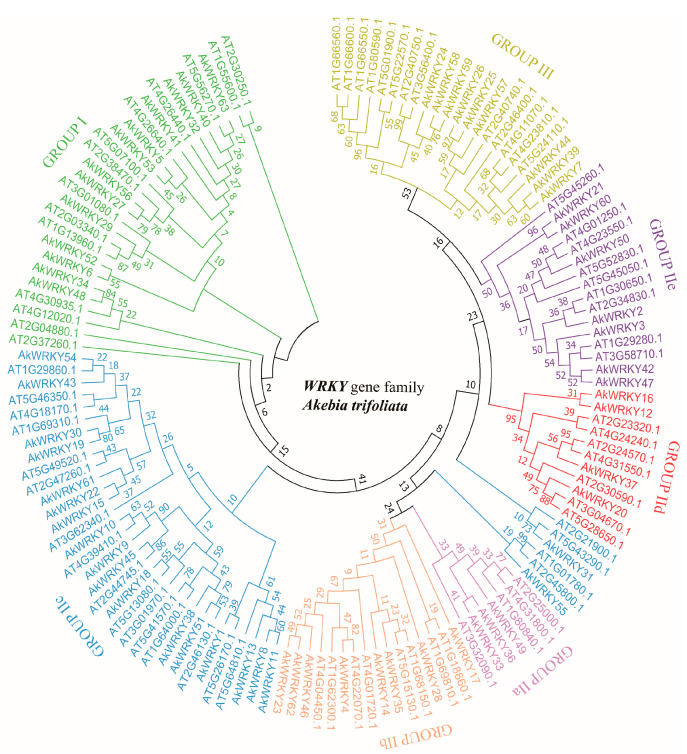
A rootless phylogenetic tree established based on the amino acid sequences of *Arabidopsis* and *A. trifoliata*. The tree divides AkWRKY proteins into seven subgroups, which are distinguished by different colors. Bootstrap confidence values from 1000 replicates are indicated at each branch.

**Figure 4 genes-13-01540-f004:**
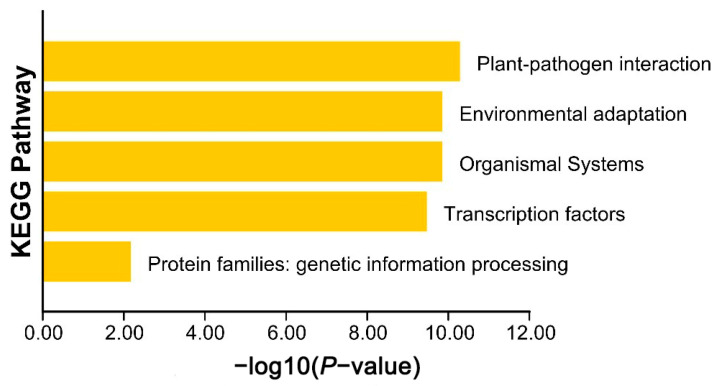
KEGG enrichment analysis map of the *WRKY* gene family. From top to bottom: plant–pathogen interactions, environmental adaptation, organic systems, TFs, and protein families (genetic information processing).

**Figure 5 genes-13-01540-f005:**
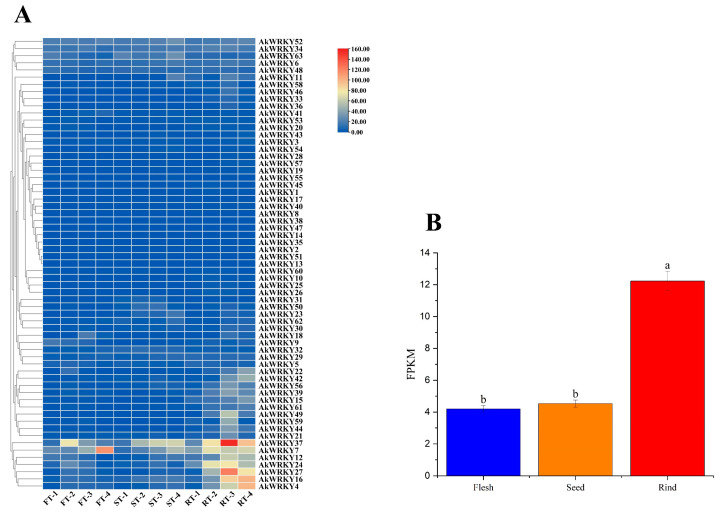
Expression analysis of the *WRKY* gene in *A. trifoliata*. (**A**) Heatmap of the expression of 63 *AkWRKY* genes at four stages in three tissues of fruit. The gradient from blue to red represents low to high expression. (**B**) Average expression of 63 *AkWRKYs* in three tissues. The different letters indicate significant differences in multiple comparisons.

**Figure 6 genes-13-01540-f006:**
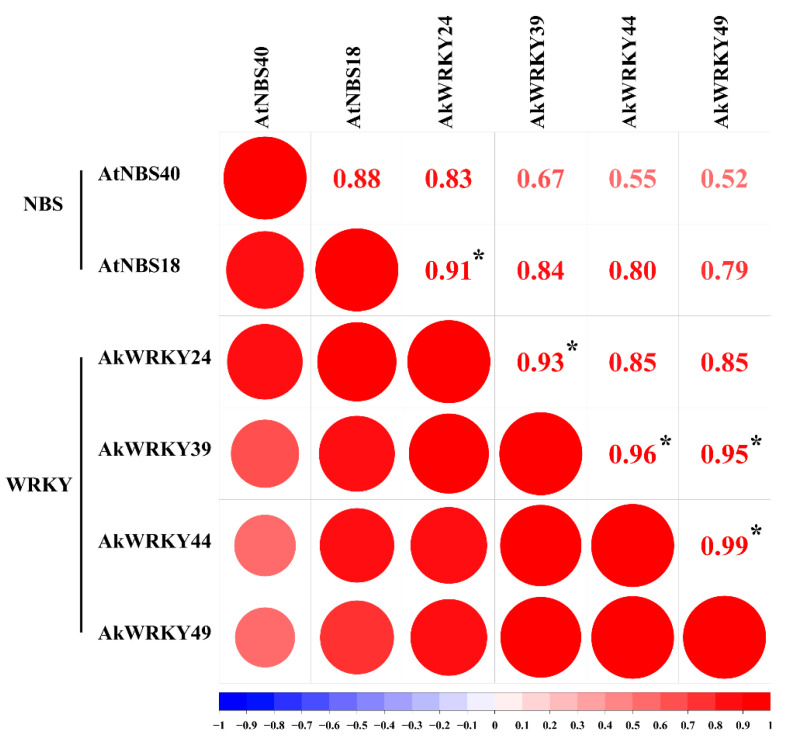
The correlation between the gene expression patterns of two *AkNBS* genes and four *AkWRKY* genes. * Indicates a significant correlation at *p* = 0.05 level.

**Table 1 genes-13-01540-t001:** Groups of *WRKY* genes in the *A. trifoliata* genome.

Type	Num.	Gene Length (Exon Number)		
		Min	Max	Maen
**Group I**	**13**	3129 (*AkWRKY63*)	12,174 (*AkWRKY48*)	6365.69 ^a^ (5.31) ^a^
**Group II**	**41**	698 (*AkWRKY55*)	7626 (*AkWRKY45*)	3368.76 ^b^ (3.63) ^b^
Subgroup IIa	3			
Subgroup IIb	8			
Subgroup IIc	19			
Subgroup IId	4			
Subgroup IIe	7			
**Group III**	**9**	1411 (*AkWRKY58*)	4013 (*AkWRKY7*)	2435.00 ^b^ (3.33) ^b^

The different letters indicate significant differences in multiple comparisons.

## Data Availability

All datasets generated in this study are presented in this paper and in the Appendix A.

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
