# Peer review of "Genome-Wide Identification and Expression Analysis of WRKY Transcription Factors in Akebiatrifoliata: A Bioinformatics Study"

_genes, 2022, doi:10.3390/genes13091540_

Round 1
Reviewer 1 Report
Dear Authors,
Reviewer comments genes-1883055
The manuscript entitled „Genome-wide identification and expression analysis of WRKY transcription factors in Akebia trifoliata“ represents a useful genome-wide bioinformatics study on WRKY transcription factors in Akebia trifoliata based on both genomic and transcriptomic data, i.e., an analysis of 12 samples including three tissues and four developmental stages by RNAseq approach. The whole study is a bioinformatics analysis using previously published data; this fact does not make the present study less valuable than an original experiment-based study; however, I think that this fact has to be clearly indicated in the manuscript title. Thus, I recommend the authors to add „a bioinformatics study“ in the manuscript title, i.e., „Genome-wide identification and expression analysis of WRKY transcription factors in Akebia trifoliata: A bioinformatics study“.
Furthermore, I have some important comments related to both methods and results which are given below:
Materials and methods, section 2.1. Data used in this study: The authors have to add the date of access to all databases and other on-line data sources used in their study since the data content in the databases can vary with time.
Materials and methods, section 2.1. The authors wrote about the transcriptomic data of 12 samples from previously done RNAseq analyses; however, no reference or web page address relevant to these RNAseq data is given. The authors have to add relevant literature reference or relevant web page / database / repository address on the transcriptomic data which they used in their study; the present manuscript cannot be accepted for publication unless these data are added!!
In Results, Figure 3, appropriate statistics has to be added to the phylogenetic tree of AkWRKY proteins, i.e., a scalebar explaining the length of the branches or numbers at nodes expressing bootstrap values per 1,000 replicates.
In Figure 6, statistical significance has to be indicated for the correlation coefficient values in the correlation matrix between four AkWRKY and two AkNBS genes expression patterns.
Formal comments:
In Intorduction, line 55, use the full plant scientific name „Akebia trifoliata“ instead of just „A. trifoliata“ when used for the first time in the text. The statement should be probably modified as follws: „Akebia trifoliata, commonly known as August melon or wild banana,….“ (not „and wild banana“).
Discussion, line 382: Add a comma between the words „and“ and „moreover“.
Conclusions, line 416: Modify the statement as follows: „The conserved zinc-finger and heptapeptide motifs were analyzed. Compared with the zinc finger structure, heptapeptide had some variations in both structure and number.“
Final recommendation: Reconsider after a major revision.
Author Response
Point 1: Materials and methods, section 2.1. Data used in this study: The authors have to add the date of access to all databases and other on-line data sources used in their study since the data content in the databases can vary with time.
Response 1: Thank you for the helpful suggestions. As for the online database access times mentioned, we have added the information in the corresponding place of the revision.
Point 2: Materials and methods, section 2.1. The authors wrote about the transcriptomic data of 12 samples from previously done RNA-seq analyses; however, no reference or web page address relevant to these RNA-seq data is given. The authors have to add relevant literature reference or relevant web page / database / repository address on the transcriptomic data which they used in their study; the present manuscript cannot be accepted for publication unless these data are added!!
Response 2: Thank you for valuable comments. In fact, the raw data of the 12 transcriptiomic data was uploaded more than one year, while the detail information was justly reported by the reference 31. As you understood, the transcriptome data used in this manuscript were all uploaded at NCBI by us. To help readers find the data and understand the resource, we had added the information such as the web page and Sequence Read Archive (SRA) number of the transcriptome in section 2.1. In addition, the corresponding reference had also been added in line 78, and the literature number is 31.
Point 3: In Results, Figure 3, appropriate statistics has to be added to the phylogenetic tree of AkWRKY proteins, i.e., a scalebar explaining the length of the branches or numbers at nodes expressing bootstrap values per 1,000 replicates.
Response 3: Sorry for the missing information. Here, we had added the information in Figure and also given the bootstrap value in the illustration of Figure 3. Please check it again.
Point 4: In Figure 6, statistical significance has to be indicated for the correlation coefficient values in the correlation matrix between four AkWRKY and two AkNBS genes expression patterns.
Response 4: Thanks for your suggestions. We have added statistical significance in Figure 6 and also added corresponding legends in the manuscript.
Point 5: In Introduction, line 55, use the full plant scientific name “Akebia trifoliata” instead of just “A. trifoliata” when used for the first time in the text. The statement should be probably modified as follows: “Akebia trifoliata, commonly known as August melon or wild banana….” (not “and wild banana”).
Response 5: Thank you for your helpful recommendation. We had modified the expression according to your recommendation.
Point 6: Discussion, line 382: Add a comma between the words “and” and “moreover”.
Response 6: Nice. We had fixed the problem.
Point 7: Conclusions, line 416: Modify the statement as follows: “The conserved zinc-finger and heptapeptide motifs were analyzed. Compared with the zinc finger structure, heptapeptide had some variations in both structure and number.”
Response 7: Ok. Your valuable suggestion had been adopted by us in the revision.
Reviewer 2 Report
In this manuscript, Zhu et al. performed a genome-wide and transcriptomic analysis of WRKY TF in A. trifoliata. In silico analysis identified 63 WRKY genes. Although the study provides some useful information on a species that is growing in interest in Asia, the study requires improvement prior to being processed further. The authors can find all my comments in the attached PDF version of their manuscript.

Author Response
Point 1: Is this work (genome and transcriptome) belong to the authors? In any case, provide a proper reference.
Response 1: Sorry for the unclear expression about the information of both genome and transcriptome. As you suspected, both the data were produced and uploaded by ourselves. In revision, we had added the information about them such as the web page, Sequence Read Archive (SRA) number and literature citation in the corresponding position of the revision.
Point 2: Add GEO and SRA numbers.
Response 2: Thank you for you helpful recommendation. This information of SRA you referred had been added in the revision, but there was not the available number of GEO number because the transcriptome data was not uploaded to the GEO database. Please check it again.
Point 3: The Transcriptome of AtWRKY genes is poorly described. More information of the genes and their putative functions is required. In addition, the authors should present differentially expressed genes and commonalities between plant tissues and discuss accordingly. The authors could list at least 5 to 10 topmost upregulated or downregulated genes.
Response 3: Thank you for your valuable suggestions. After carefully consider your comments, we had added a description of top differential expression genes and some discussions in 3.5 and 4.5, respectively, according to your suggestions. Please review it again.
Point 4: This section can be improved after adding additional data on Transcriptome (section 4.4).
Response 4: Ok. We had added transcriptomic data and corresponding discussions about W-box, which reinforced our opinion and enhanced the logical expression. Please give some helps if there is still the addition suggestion.
Point 5: The conclusion can be improved after improving the results and discussion sections.
Response 5: Thank you useful comments. We had improved the conclusion part according to your suggestion. As far as the change details were concerned, please see the revision.
In addition, we have carefully revised questions raised by reviewers regarding abbreviations, introductory sentences, and reference format. For example, A. trifoliata in the abstract was changed to Akebia trifoliata, and the corresponding doi number was added after the reference.
Round 2
Reviewer 1 Report
Dear Authors,
I appreciate your response to all my comments on your manuscript and the manuscript modification accordng to my comments.
I have no further comments on the revised manuscript and I can recommend publication of the revised manuscript in the present form.
Reviewer 2 Report
The authors have improved the manuscript according to the Reviewer's comments.